# A million shades of green: understanding and harnessing plant metabolic diversity

Rocky D Payet [ID][1✉], Adnane Aouidate[1,2], Rebecca Casson[1], Alan Houghton[1], Mai-Truc Pham [ID][1] &
Anne Osbourn [ID][1✉]

## Abstract

**Recent developments in single-cell -omic and metabolite imaging technologies and the increasing availability of high-quality genome assemblies are having a transformative impact on the way research is carried out into plant specialised metabolism. Integrating these technologies into pathway discovery projects is therefore highly advantageous. Here, we present a general introduction into methods and workflows in specialised metabolism research. We review a range of recent methodologies, highlighting what they might be used for and common pitfalls which may be encountered. Finally, we provide a practical guide on how these technologies may be incorporated into a specialised metabolic pathway discovery pipeline for researchers who are new to the field.**

**Keywords** Natural Products; Plant Metabolism; Specialised Metabolism
**Subject Categories** Metabolism; Plant Biology

## Introduction

Plants produce a wide diversity of compounds. Broadly, these are separated into primary metabolites, which are necessary for growth and development, and specialised metabolites (sometimes called natural products, see "Glossary"), which serve a host of ecological functions ranging from defence, communication with other organisms, and interaction with their environment (Suresh et al, 2023). It should be noted, however, that the distinction between these two groups of compounds is not always well defined (Ji et al, 2024), such as in the case of sterols. Humans have exploited natural products for over 5000 years (Petrovska, 2012). More than 30% of our drugs derive from a direct plant source, and more than 60% of small molecule drugs introduced in the past 20 years are based on bioactive molecules from plant extracts or their derivatives. To date, over 200,000 plant natural products have been reported, yet genomic predictions suggest that plants can make millions of structurally varied molecules (Fang et al, 2019),

indicating there are many more still to discover. The complexities of plant genomes, such as size and high repeat content, have traditionally presented substantial challenges to elucidation of the genes and enzymes of plant natural product biosynthetic pathways. However, the advent of long read sequencing technologies such as single-molecule real-time (SMRT/PacBio) sequencing and the reduced cost of sequencing has helped to overcome these barriers (van Dijk et al, 2018), heralding a new era of high-quality plant genome assemblies (Marks et al, 2021). Natural products (and indeed research into metabolism in general) research is an exciting, fast-moving area, but the multifaceted and interdisciplinary nature of the subject makes it a large and complex field. In this article, we attempt to demystify natural products research by outlining the archetypal steps to elucidate a biosynthesis pathway, while addressing the key considerations and common challenges faced. These steps encompass determining a product of interest or species of interest to work on, identifying candidate biosynthetic genes and pathways, and elucidating the activities of these in a heterologous host. We emphasise recent technological advancements in the specialised metabolism space and outline key practical considerations for those who are unfamiliar with the technologies, or how they may be incorporated into a gene discovery pipeline.

## Augmenting natural products research through the use of cheminformatics databases

A common entry point to natural products research is through an activity-guided, empirical approach in which an unknown compound with known (and desired) activity is tracked and isolated using various fractionation experiments (Nothias et al, 2018). These methods are often highly successful but require analytical chemistry infrastructure to isolate and assign structures to compounds, as well as a large quantity of starting material. This approach is still employed, but technological advances have put emphasis on high-throughput methods (Ayon, 2023), with particular focus on automated screening of vast compound libraries—a process often referred to as bioprospecting. There are about 391,000 species of plant known to science at the time of writing this article, and as many as 2000 new plant species are described every year (Antonelli et al, 2023). Prioritisation of plant species to focus on for particular purposes has often been guided by ethnobotany (Teixidor-Toneu et al, 2018).

---

[1]John Innes Centre, Norwich Research Park, Norwich, UK. [2]School of Applied Sciences - Ait Melloul, University Ibn Zohr, Agadir, Morocco. ✉E-mail: Rocky.payet@jic.ac.uk;
Anne.osbourn@jic.ac.uk

**Glossary**

**Natural products**

Specialised metabolites produced by living organisms, typically excluding large biomolecules such as proteins and nucleic acids.

**Cheminformatics**

Cheminformatics (also known as chemical informatics) uses computational methods to address chemical challenges and derive insights from chemical data. It encompasses database management, data mining, visualisation, model development, application development, property prediction, etc. Relevant chemical data includes small-molecule formulas, structures, properties, spectra, as well as biological and industrial activities.

**Box 1  Chemical databases**

Reaxys: https://www.reaxys.com/#/search/quick/query. A database that contains millions of chemical substances and reactions, along with their associated chemical and physical properties, as well as literature references.

Lotus: https://lotus.naturalproducts.net/. One of the largest and best-annotated resources for NP occurrences, available free of charge and without any restrictions. LOTUS is a dynamic database that is hosted both on its official website and on Wikidata. The Wikidata version enables community curation and the addition of new data.

NPASS: https://bidd.group/NPASS/. Natural Product Activity and Species Source is designed to provide a freely accessible database that integrates detailed information on species, sources, and biological activities of natural products.

Coconut: https://coconut.naturalproducts.net/. The COlleCtion of Open NatUral producTs is a platform that supports natural product research by offering data, tools, and services for deposition, curation, and reuse. Launched in 2021, it has become one of the largest open natural product databases. COCONUT includes chemical structures, names and synonyms, species, organism parts, geographic information on sample collection locations, and available literature references.

CheMBL: https://www.ebi.ac.uk/chembl/. A manually curated database of bioactive molecules with drug-like properties. This database combines chemical, bioactivity, and genomic data to facilitate the translation of genomic information into effective new drugs.

In situations where one has a given bioactivity or property of interest (e.g. anti-fungal or anti-cancer activity), or perhaps a common molecular feature associated with a desired property (e.g. natural products which derive from 3-hydroxyflavone, the backbone of all flavonols), it may be appropriate to consult specialised databases, such as Coconut (Sorokina et al, 2021), Lotus (Rutz et al, 2022) or NPASS (Zhao et al, 2023). These databases compile information from globally published research, concentrating on natural products and metabolites that have been extracted and identified using techniques such as high-performance liquid chromatography (HPLC), gas chromatography-mass spectrometry (GC-MS), liquid chromatography-mass spectrometry (LC-MS), and advanced nuclear magnetic resonance (NMR) spectroscopy. Furthermore, it is possible to filter these databases based on molecular features, species/genera of origin, and/or bioactivities using varied, recent approaches (Boldini et al, 2024; Gaudry et al, 2024). General databases such as Reaxys® (Elsevier Limited; https://www.reaxys.com/#/search/quick/query) or ChEMBL (Mendez et al, 2019), which are not restricted to natural products or bioactive molecules, may also be consulted (Box 1). It is, however, essential to recognise that data within these databases are derived from published studies that may contain inaccuracies, such as incorrect stereochemistry assignments, typographical errors in species or genus names, or misidentification of compounds. To mitigate these issues, it is advisable to first select only those molecules with complete stereochemistry, followed by thorough manual verification to ensure accurate structural assignment. Additionally, cross-referencing with taxonomic databases such as the National Center for Biotechnology Information taxonomy database (Schoch et al, 2020) can help confirm the accuracy of species names. Errors may also arise in cases in which the organism of study has not been properly taxonomically identified. This is particularly difficult to resolve in studies that lack sufficient metadata or information on provenance, as it can lead to misassumptions.

Curated and filtered data from these databases can be valuable for various purposes. For example, surveying the occurrence of a given compound or core structure across different plant lineages can provide insights into the underpinning evolutionary dynamics. It can also be used to inform on the likely phytochemistry of species which have not yet been the subject of study, based on their relationship to other known species (Rodríguez-López et al, 2022). In other scenarios, one may have a compound of interest and wish to determine how it is biosynthesised in a particular plant species. Here, chemical databases may also be useful because they can provide information on other plant species that may produce the compound of interest, and/or shed light on related compounds that are synthesised by the same species.

## Identifying biosynthetic gene candidates

Once a target compound and/or plant species for investigation has been identified, it is next ideal to obtain material of that species. This can later be used to gather important information about where to look for biosynthesis genes and can also be used to clone gene candidates directly, which is cheaper than DNA synthesis. This material may be obtained through a botanical garden or a commercial supplier such as a plant nursery. Whichever route is taken, it is important to secure information about the provenance of the material in order to ensure Nagoya Protocol compliancy (Aubertin and Filoche, 2011). This can be challenging in many countries, where established processes are lacking. Smith et al (2018) provide comprehensive guidelines on navigating Nagoya and are a good first point of call. The metabolite profiles of commercially purchased plant material may differ substantially from bona fide botanic garden accessions, possibly due to nomenclature issues or hybridisation with other closely related species. Indeed, even within the same species it is well documented that different chemotypes can exist between individuals, and thus the specimen of study may not mirror published data (Anaia et al, 2024; Ziaja and Müller, 2025). It can be a good idea to prepare herbarium specimens of the material used. This is particularly salient for groups of species which lack genomic information, and can serve to ensure reproducibility and traceability of subsequent work (Davis and Knapp, 2024). The plant material obtained may be clonally propagated, e.g. cuttings or bulbs. However, in some instances the only material available is seed, and in this scenario ensuring successful germination can be an issue.

Once material from the plant of interest has been obtained, the next step is to confirm the presence of the natural product of interest, where it accumulates, and, ideally, in which tissue it is biosynthesised—this is a vital piece of information to guide later gene discovery. Classically, determining sites of accumulation would entail extraction from a tissue homogenate followed by analysis by LC-MS or GC-MS. More recently, matrix-assisted laser desorption/ionisation (MALDI) imaging approaches (which do not involve homogenisation) have been used to investigate the localisation of target compounds at the cell type level, thus greatly improving the resolution of these endeavours. In these methods, plant material is finely sliced (e.g. to 80 μm thickness (Yamamoto et al, 2019)) and mounted on a glass slide. The sample is then freeze-dried, immersed in a matrix solution to facilitate ionisation, and then subjected to MALDI imaging spectrometry (Nakabayashi et al, 2021; Yamamoto et al, 2019). This has allowed for localisation within tissues (Nakabayashi et al, 2021) and even at the level of specific cell types within tissues (Yamamoto et al, 2019), down to a resolution of 10 μm. While MALDI imaging is conceptually straightforward, the selection of an appropriate matrix for a tissue of interest can be empirical, potentially requiring a number of different options to be explored (Alolga et al, 2024). Additionally, it does rely heavily on tissue sectioning, which may be difficult in tissues which are very small or difficult to work with.

In cases where the biosynthesis of a natural product of interest is conditional, this feature can be exploited to facilitate pathway discovery, e.g. hormone elicitors (Šenkyřík et al, 2023) or biotic factors (Katoch et al, 2022) can greatly stimulate the biosynthesis of some natural products. To determine sites of biosynthesis, labelling experiments that measure the incorporation of labelled radio-isotopes or stable isotopes into final product remain the "gold standard" approach (Eljounaidi et al, 2024; Mehta et al, 2024; Trojanowska et al, 2000). These approaches can also be combined with MALDI imaging (Schwaiger-Haber et al, 2023), allowing for highly precise determination of sites of synthesis at the cell type level. Furthermore, knowledge of the precise localisation of a given compound in a native tissue can suggest potential function and can yield insights into how precursors are partitioned to facilitate flux.

Having established which tissues/conditions the natural product of interest is located/biosynthesised in, the next step in the pathway elucidation process is to access sequence information. Next-generation sequencing (NGS) resources pertaining to the plant of study may already be publicly available; the 1KP project is an excellent first port of call for this, containing NGS data for over 1000 species of plant (Carpenter et al, 2019). However, issues with sequencing depth can be encountered with this database, which can complicate later cloning efforts. Furthermore, these transcriptomes typically consist of one organ per species or of homogenised seedlings and therefore may not span sites of biosynthesis. Finally, the sequenced individual in the 1KP database may not well represent the accession acquired, for example in species which show high levels of intraspecific variation. It is therefore preferable to generate in-house NGS resources from the accession in use. This may involve performing RNAseq on different tissues, followed by either mapping back to a published transcriptome, or more commonly generation of de novo transcriptome resources for different plant tissues/treatment conditions. Typically, this would involve short-read Illumina sequencing. However, more recently long-read Iso-Seq has been employed in the construction of

reference-quality transcriptomes (Zhang et al, 2022). While transcriptomes can be obtained from a single library, it is advisable to design a sequencing experiment to cover a range of tissues in which biosynthesis is and is not taking place, as this allows for exclusion of genes that are unlikely to be involved in the pathway of interest. Furthermore, quantitative information is often a useful line of evidence in pathway discovery: for example, genes that have very low expression may be excluded, and genes that are differentially expressed in tissues where biosynthesis is taking place can be shortlisted. It is therefore recommended that a minimum of three replicates per tissue, preferably four, are sequenced, as this allows for robust, statistical quantification of transcript abundance.

Single-cell sequencing technologies have also recently been developed, and have considerable potential for facilitating elucidation of natural product pathways that are restricted to specific cell types (Lin et al, 2023; Wang et al, 2022), as well as examples of pathways which are split between multiple, different cell types (Ozber and Facchini, 2022; Sun et al, 2023). These sequencing methods rely on the dissociation of tissues into cells in suspension, followed by preparation of protoplasts (Cole et al, 2021). The suspended protoplasts are then attached onto individually barcoded particles using microfluidics, followed by preparation of libraries and sequencing. The data can then be resolved into distinct cell types through use of principle component clustering (Cole et al, 2021). It should be noted, however, that the preparation of protoplasts for some tissues and some species can be extremely challenging. In such cases, single nucleus sequencing might be more feasible. The workflow for single nucleus sequencing is similar to single cell, but crucially it does not rely on the preparation of protoplasts (Sunaga-Franze et al, 2021). These methods are contrasted in Box 2, and well-reviewed by Ding et al (2020). These single cell and nucleus sequencing methods are substantially different from bulk RNA-seq, in which whole tissues are homogenised, as the information on which cells are expressing which genes is lost. Furthermore, transcript abundance in bulk RNA-seq is a synthesis of level of expression and abundance of cell type in which expression is taking place. Single-cell sequencing circumvents this problem, however in both methods the architecture of where genes are expressed in the context of a tissue is lost. Very recently, spatial transcriptomic methods such as stereo-seq have been developed and incorporated into plant science, which can overcome both these issues (Yin et al, 2023).

While many of the provided examples of single-cell sequencing use Illumina short-read sequencing technologies, the decreasing costs of SMRT sequencing have led to more widespread adoption (van Dijk et al, 2018). This long-read sequencing type permits differentiation between transcript isoforms and has recently been integrated into single-cell (Al'Khafaji et al, 2024; Shi et al, 2023) and even single-nucleus sequencing pipelines (Hardwick et al, 2022). However, it should be noted that there exist examples of biosynthetic pathways for which the functions are split over multiple different cell types, such as monoterpene indole alkaloids in *Catharanthus roseus* (Li et al, 2023) and morphine biosynthesis in *Papaver somniferum* (Ozber and Facchini, 2022). Sequencing or focusing on a single cell-type may therefore not always be appropriate. A comparison between long-read and short-read sequencing technologies is made in Box 2.

In addition to transcriptome sequencing applications, SMRT sequencing has had a transformative impact on genomics, resulting

**Box 2.  NGS sequencing methods and their uses**

| Short-read sequencing | | Long-read sequencing | |
|---|---|---|---|
| **Pros** | **Cons** | **Pros** | **Cons** |
| Cheap | Difficult to resolve highly repetitive/complex sequences | Can resolve highly repetitive/complex sequences | More expensive |
| Can achieve high sequencing depth | Cannot be used to distinguish sequence isoforms well | Can be used to distinguish isoforms | Lower accuracy per read |
| Ideal for standard transcriptomics | | Ideal for reference transcriptomes/ genomes | |
| **Single-cell sequencing** | | **Single nucleus sequencing** | |
| **Pros** | **Cons** | **Pros** | **Cons** |
| Isolated cells can be used for further multi-omics (proteomics/ metabolomics) | Rely on generating protoplasts for cell sorting | Does not rely on generating protoplasts | Cannot be used to profile proteome or metabolome |
| Transcripts across the cell are represented (e.g. in the cytoplasm) | Ease of dissociation of cells can introduce heavy bias | Reduced bias of ease of dissociation | Only captures transcripts in nuclei |
| | Cannot use with frozen samples | Can be used with frozen samples. | |

in a dramatic increase in the number of sequenced plant genomes. This can be combined with chromosome conformation capture technologies (Hi-C), which reveal contacts between genomic regions which are in close three-dimensional proximity (Šimková et al, 2024), to resolve even very complex genomes down to chromosome-level assemblies (Chávez Montes et al, 2022). These developments have led to the discovery that in a burgeoning number of cases, genes involved in particular natural product biosynthetic pathways are co-located in the genome in biosynthetic gene clusters (BGCs) (Nützmann et al, 2018; Smit and Lichman, 2022). This phenomenon offers opportunities to identify biosynthetic pathway genes (Kerwin et al, 2024; Liu et al, 2020; Winzer et al, 2012) based on genomic location, which can greatly accelerate the pathway elucidation process. Such clusters can readily be predicted by publicly available algorithms, such as plantiSMASH (Kautsar et al, 2017). Importantly, clustering can enable the discovery of genes encoding unexpected pathway components that would not have been found otherwise, based on a Rosetta Stone approach, e.g. a reductase involved in the D-fucosylation of the triterpene glycoside adjuvant QS-21 (Reed et al, 2023). Key considerations to take into account when sequencing a plant genome are the ploidy of the organism and its heterozygosity, and sequencing depth should be adjusted accordingly.

Knowledge of precursor molecules allows for probable routes of biosynthesis to be inferred. While natural products are highly diverse, many enzymatic transformations are common to most classes. For example, hydroxylations around a carbon skeleton are frequently made by cytochrome P450 family members (Nguyen and Dang, 2021). This allows gene lists to be curated for likely relevant classes of enzyme. However, in cases when a non-canonical gene family is involved in a pathway, or if there is no precedent for the step in literature, then difficulties may be encountered. If a pathway is clustered in a BGC, then examining other genes in the cluster may lead to discovery of missing steps, for example to a scaffold protein required for the formation of a protein complex required for biosynthetic pathway function (Boccia et al, 2024; Jozwiak et al,

2024). Where pathway genes are not organised in BGCs, differential gene expression analyses between tissues/conditions where the metabolite is accumulated to high or low levels (Carroll et al, 2023; Payet et al, 2024), or correlation with known biosynthesis enzymes (Jo et al, 2024) may all help identify functional genes.

Finally, there is a current drive to develop gene discovery pipelines which incorporate multiple different omics strategies—typically genomics, transcriptomics and metabolomics (Louwen et al, 2023). These multi-omic methods have historically been applied to microbial systems but are successfully being implemented in plant systems (Li et al, 2023). These approaches are particularly powerful, as they incorporate multiple lines of evidence. It should be noted, however, that to achieve sufficient statistical power it is generally necessary to sample multiple tissues/ conditions in replicate, with a minimum of three, preferably four replicates per sample; however this can be expensive. Moreover, such an experiment should be designed to span both tissues/ conditions where biosynthesis is known or suspected and tissues/ conditions where it is not, so that meaningful comparisons can be made.

# Ratification of candidate genes and pathways

Following the identification of candidate genes, the next step in the process is to validate their function. This process can be broadly categorised into two different approaches: "bottom-up" and "top-down".

In the "bottom-up" approach, candidate genes are expressed in a heterologous host. This host may be microbial, e.g. yeast or bacteria, or multicellular, i.e. plants. Microbial systems are often tractable and easy to scale up—for example, *Escherichia coli* is readily transformable and can be grown to high density in a controlled incubator. However, some of the key enzyme families

involved in natural product biosynthesis in plants, such as cytochrome P450s, are localised on the endoplasmic reticulum, an internal structure that bacteria lack. Eukaryotic microbes such as yeast can provide useful alternatives for expression and purification of such enzymes for expression of individual steps (Nguyen et al, 2023) or indeed whole pathways (Winegar et al, 2024). However, these typically require additional metabolic engineering to provide the necessary precursors and co-factors. The plant system *Nicotiana benthamiana* (a wild relative of tobacco) is emerging as a rapid and powerful alternative to these microbial systems (Golubova et al, 2024), as it is better suited for expression of plant natural product enzymes and will naturally produce most of the necessary co-factors. Furthermore, in addition to stable transformation, *N. benthamiana* is amenable to transient transformation by agroinfiltration (Sainsbury et al, 2012), allowing for rapid screening of putative biosynthesis genes (Carlson et al, 2023; Reed et al, 2017).

The first step of the "bottom-up" approach is to clone the candidate gene into a suitable expression vector, such as pEAQ (Sainsbury et al, 2009). This expression cassette is then transformed into *Agrobacterium tumefaciens*, and *A. tumefaciens* cells bearing the expression constructs are then infiltrated into the leaves of *N. benthamiana* (Sainsbury et al, 2012). Expression of the introduced genes peaks after 4–5 days (Reed et al, 2017), after which plant material is then harvested, the metabolites extracted into an appropriate solvent, and either LC-MS or GC-MS is used to determine the presence of the target metabolite. Successive genes can be stacked together, either in multi-gene constructs (Payet et al, 2024) or by combining agrobacterium strains (Reed et al, 2017), and thus complete biosynthetic pathways can be reconstituted.

One common issue experienced with this approach is that precursor pools may be limiting and unable to meet the demand of constitutively expressed downstream enzymes. This can be addressed by boosting precursor supply, for example by co-expressing a feedback insensitive truncated 3-hydroxy-3-methyl-glutaryl-CoA reductase gene to boost triterpene production (Reed et al, 2017; Rodríguez-Concepción and Boronat, 2015).

When the host species is ameliorable to transformation, a "top-down" approach can be used to validate candidate gene function (Small, 2007). In this approach, small interfering RNAs can be expressed to activate endogenous gene silencing mechanisms, thereby reducing expression of the candidate gene and, correspondingly, production of the target molecule (Boccia et al, 2024; Jo et al, 2024). While this is an effective strategy, it should be noted that typically RNAi methods do not achieve complete silencing, and so this method rarely results in a complete knockout phenotype. Furthermore, silencing is often transient, and over time the efficacy may reduce. CRISPR-Cas can also be deployed to manipulate gene expression, either by silencing using a nuclease dead Cas9 (Zhang et al, 2023), or to knock out genes at a genetic level (Mercx et al, 2017). Biosynthetic genes can also be overexpressed in such systems to further demonstrate their involvement in planta (Grzech et al, 2024). Additionally, depending on whether the metabolites are present in root tissue, transformation via *Agrobacterium rhizogenes* to generate hairy roots can provide an excellent platform to test candidate genes (Shi et al, 2021). These stably transformed roots can be propagated indefinitely, providing a potential reservoir of pathway intermediates that can be isolated and used for in vitro enzyme assays. Furthermore, hairy roots are routinely genetically manipulated by CRISPR-Cas systems, allowing for complete knockout of candidate genes (Kiryushkin et al, 2021).

Generally, the "bottom-up" approach is faster and more accessible, as it does not require the organism of study to be transformed. However, the "top-down" method has many advantages in certain situations. For example, for many years steroidal glycoalkaloids (SGA) from potato and tomato could not be reconstituted in a heterologous host as the recently described scaffold protein *GAME15* was not known (Boccia et al, 2024; Jozwiak et al, 2024). As such, many SGA pathway steps were discovered based on a "top-down" strategy, in which genes were knocked out or overexpressed in the native organism (Sonawane et al, 2022; Nakayasu et al, 2021). The "top-down" approach also serves as a formal validation of function in the producing plant, whereas this information is lost in a "bottom-up" approach.

## Detection and structural assignment of natural products

Compounds produced by heterologous expression are typically validated through mass spectrometry coupled chromatographic methods such as GC-MS and LC-MS. This allows for the identification of the putative products through their molecular mass and retention times. Furthermore, through comparison of the fragmentation pattern of known, similar compounds (such as those of a closely related standard) it is possible to suggest potential structures of unknown products based on the fragments observed and some knowledge of the likely activity of the tested genes (Moses et al, 2014b).

Additionally, the use of spectral libraries can assist with the identification of products if samples are run under identical conditions (Morehouse et al, 2023). It should be noted, however, that unambiguous identification requires either the exact matching of retention time and mass spectrum to an authenticated standard, or else the purification and structural elucidation of the product through other means, as mass spectrometry cannot unambiguously determine structure by itself due to structural isomers (and especially enantiomers) commonly sharing identical mass spectra (Zhou et al, 2022).

To characterise the structure of unknown products from heterologous expression experiments, higher quantities of product are needed. To accommodate this, infiltrations are typically scaled up for metabolite extraction and isolation (Reed et al, 2017). Given the complex nature of plant material, extraction and isolation of pure compounds necessitates various separation techniques and strategies. These techniques include critical sample preparation steps such as drying, grinding, decolorising, defatting, and fractionating of crude extracts prior to column chromatography. Advances in modern extraction techniques, such as pressurised liquid extraction and solid-phase extraction, have enhanced sample extraction efficiency, complementing conventional methods such as soaking, Soxhlet extraction, and ultrasound-assisted extraction (Cheok et al, 2014; Majinda, 2012). Furthermore, alongside flash chromatography and medium-pressure LC for rapid preliminary fractionation of complex mixtures, HPLC is regarded as a highly effective and convenient technique for compound isolation. HPLC systems can be coupled with various types of detectors, such as mass selective detectors, photodiode array detectors or evaporative

light scattering detectors, facilitating the handling of non-ultraviolet absorbing compounds (Bucar et al, 2013; Sticher, 2008).

**Box 3   NMR methods and their uses**

2D NMR experiments

   COSY: (correlation spectroscopy) cross-peaks show signals of protons attached to carbons that are adjacent to each other.

   TOCSY: (total correlation spectroscopy) is much like COSY, except can show longer-range neighbouring protons within a group of connected carbons (or spin system).

   NOESY: (nuclear Overhauser effect spectroscopy) and ROESY (rotating-frame nuclear Overhauser effect spectroscopy) show cross-peaks when protons are spatially close to each other. These experiments are particularly useful for determining stereochemistry.

   HSQC: (heteronuclear single quantum coherence) shows which protons are directly attached to which carbon.

   HMBC: (homonuclear multiple bond coherence) shows connections between carbons and protons separated by two, three or sometimes even four bonds. These cross peaks can be used to assign carbon-carbon connectivity.

   HSQC-TOCSY: (heteronuclear single quantum coherence-total correlation) looks like a normal TOCSY but is additionally resolved using the $^{13}C$ dimension, which can be useful for resolving overlapping cross-peaks. This can be particularly useful with molecules that contain multiple sugars, where the cross-peaks would normally overlap in a standard TOCSY experiment.

   H2BC: (homonuclear 2 bond coherence) is similar to HMBC, except that the cross-peaks are limited to a distance of two bonds. This spectrum is usually used to distinguish the carbons directly adjacent to the carbon to which the proton is attached.

**Box 4.   Take home messages**

- Cheminformatics databases can be used to systematically identify compounds based on structural features (e.g. shared aglycone) and bioactivity and can also be integrated with taxonomy to investigate chemistry across genera.
- Understanding where and when biosynthesis occurs is vital for metabolic pathway elucidation.
- Single-cell omics techniques are powerful additions to pathway elucidation, but it must be taken into consideration that some pathways are segmented over a variety of different cell types.
- Heterologous reconstitution of biosynthesis pathways in *Nicotiana benthamiana* is effective and highly scalable.

The structural elucidation of natural products primarily relies on the use of advanced NMR spectroscopy (Bross-Walch et al, 2005; Kwan and Huang, 2008). Using triterpenoids as an example, the core aglycone structure is usually characterised first through proton ($^1$H) and carbon ($^{13}$C) NMR spectra—these can be compared with published literature. In addition to discerning the core aglycone structure, the splitting pattern and chemical shift of proton signals influenced by substituents can also inform on the presence of attached moieties. For instance, resonance signals of H–29 and H–30 in the ursane scaffold appear as doublet peaks while those in the oleanane-type structure exhibit singlet peaks (Kaweetripob et al, 2016; Moses et al, 2014a; Wu et al, 2020).

Further confirmation of proposed structures is achieved using other 2D NMR techniques such as heteronuclear single quantum correlation (HSQC), heteronuclear multiple bond correlation

(HMBC), rotating-frame nuclear Overhauser effect spectroscopy, and nuclear Overhauser effect spectroscopy (Reynolds and Mazzola, 2015). However, in the case of complex, highly decorated natural products, such as saponins, obtaining comprehensive structural information—including sugar ring conformation and anomeric configuration, linkage positions and sequence of the sugar chain, as well as characterisation of acyl moieties and their linkage positions—presents significant challenges. Therefore, a combination of various 1D and 2D NMR experiments is necessary (da Silva et al, 2016).

More recently, the combined use of heteronuclear single quantum correlation-total correlation spectroscopy (HSQC-TOCSY) and heteronuclear two-bond correlation has proven beneficial for assigning carbon and proton resonances within sugar moieties (Graziani et al, 2018; Shiomi et al, 2016; Wallace et al, 2022) (Box 3).

# Future perspectives

Developments in NGS methods and metabolite imaging have greatly enriched the capacity for researchers to identify putative biosynthesis genes from a wide variety of non-reference organisms. However, one of the key bottlenecks in natural product discovery remains the ratification of biosynthesis genes. Increasing the accuracy of prediction of biosynthesis genes will therefore be vital in overcoming this. In the future, the development of large databases of ratified functional genes that can be used to train machine learning algorithms will greatly accelerate this capability.

Looking forward, the further development of multi-omic pipelines (Li et al, 2023) coupled with social network endeavours such as GNPS (Aron et al, 2020) will likely dramatically improve our capacity to identify natural product biosynthetic genes and pathways. Additionally, initiatives such as the Earth Biogenome project (Lewin et al, 2022), which aim to sequence the genomes of every eukaryote on the planet, have and will continue to revolutionise natural products research. Furthermore, advances in our understanding of protein structure and enzyme superfamilies will increase our capacity to make predictions about what biosynthesis genes may do based on their translated DNA sequences (Bordin et al, 2024). Indeed, when coupled with modern machine learning approaches, a future may be imagined where it is possible to design our own enzymes for biosynthesis of new-to-nature products based on these knowledge foundations (Notin et al, 2024) (Box 4).

Taken together, it becomes clear that the process of elucidating a natural product biosynthesis pathway is a highly multidisciplinary endeavour. It relies on skillsets in chemi- and bioinformatics to identify compounds with promising bioactivity and putative biosynthesis genes, molecular biology to clone and perform heterologous expression experiments, and chemistry to purify and assign structures to newly produced compounds (Fig. 1). As the techniques employed in each of the described steps become increasingly advanced, so too will the need to employ multidisciplinary teams to address the challenges of natural products research. The curation of these highly interdisciplinary teams also provides excellent training opportunities for early career researchers.

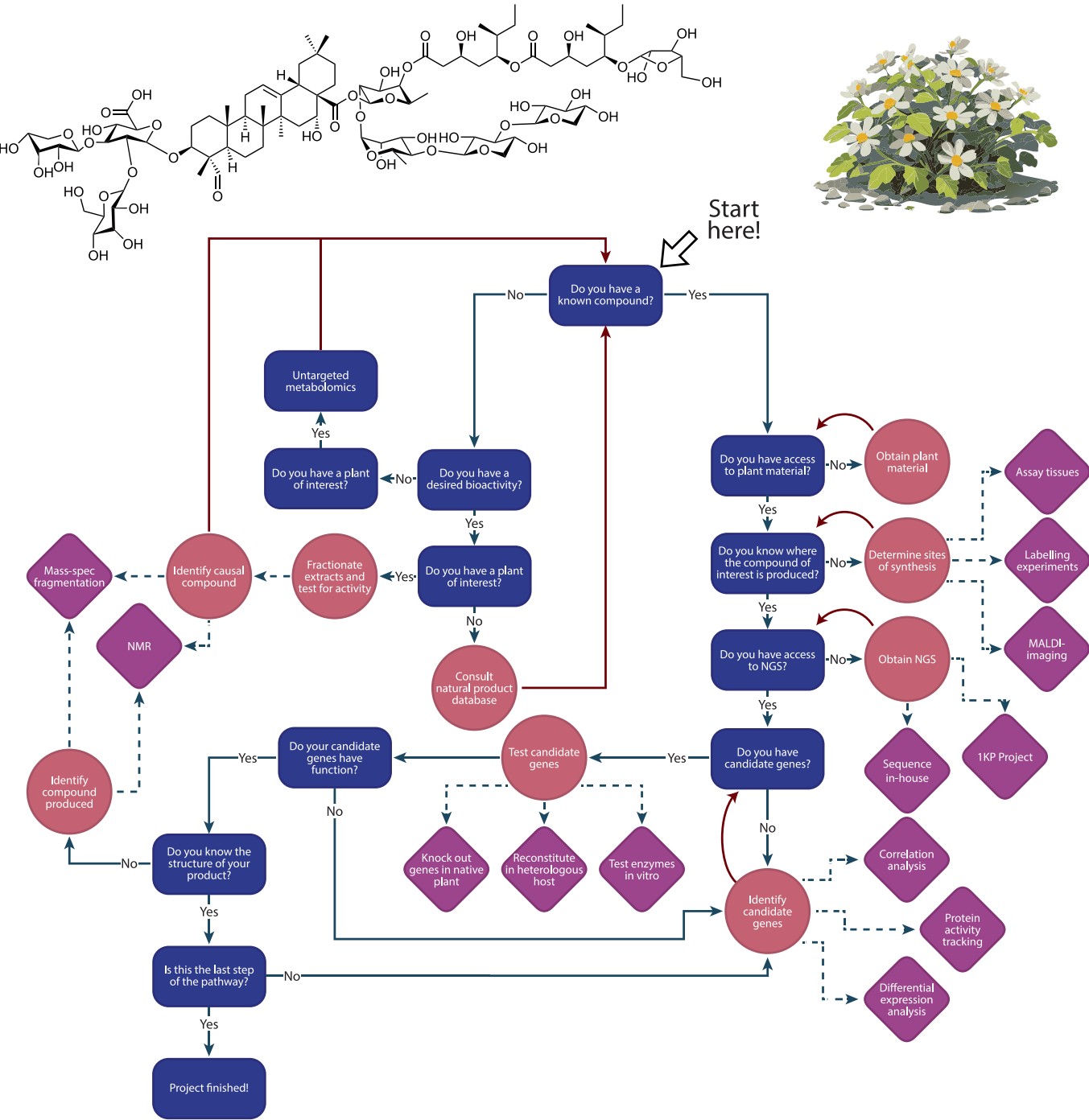

**Figure 1.   Flow chart showing the key stages of a specialised metabolite gene discovery pipeline.**

Rectangular boxes, key milestones; circles, action to take at the key milestones; diamonds with dashed arrows, specific techniques by which the circles would be achieved. Readers may wish to align their current research project with this chart to decide potential next steps.

# Peer review information

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

## Acknowledgements

We would like to acknowledge the following funding sources to the Osbourn lab: The Novozymes Prize 2023 (Novo Nordisk Foundation), Wellcome Discovery Award #227375/Z/23/Z, BBSRC responsive mode award APP3941, the BBRSC Institute Strategic Programme Grant "Harnessing biosynthesis for sustainable food and health" (BB/X01097X/1) and the John Innes Foundation.

## Author contributions

**Rocky D Payet**: Conceptualisation; Writing—original draft; Writing—review and editing. **Adnane Aouidate**: Writing—original draft. **Rebecca Casson**: Writing—original draft. **Alan Houghton**: Writing—original draft. **Mai-Truc Pham**: Writing—original draft. **Anne Osbourn**: Conceptualisation; Supervision; Funding acquisition; Project administration; Writing—review and editing.

## Disclosure and competing interests statement

