## [Peer Review File · The EMBO Journal]

A million shades of green: Understanding and harnessing plant metabolic diversity

Rocky Payet, Adnane Aouidate, Rebecca Casson, Alan Houghton, Mai-Truc Pham, and Anne Osbourn

Corresponding author(s): Rocky Payet (rocky.payet@jic.ac.uk) , Anne Osbourn (Anne.Osbourn@jic.ac.uk)

Review Timeline:

Submission Date:	21st Feb 25
Editorial Decision:	24th Mar 25
Revision Received:	19th May 25
Accepted:	2nd Jun 25

Editor: William Teale

Transaction Report:

Dear Rocky,

Thank you for submitting your manuscript for consideration by the EMBO Journal. It has now been seen by two referees whose comments are enclosed. As you will see, both are broadly in favour of publication, pending satisfactory minor revisions.

Given the referees' positive recommendations, I would like to invite you to submit a revised version of the manuscript, addressing the comments of all three reviewers.

Thank you for the opportunity to consider your work for publication. I look forward to your revision.

Best wishes,

William

William Teale, PhD
Editor
The EMBO Journal
w.teale@embojournal.org

We realize that it is difficult to revise to a specific deadline. In the interest of protecting the conceptual advance provided by the work, we recommend a revision within 3 months (22nd Jun 2025). Please discuss the revision progress ahead of this time with the editor if you require more time to complete the revisions. Use the link below to submit your revision:

Referee #1:

In this manuscript, Payet et al. present an extensive overview of the latest technological advances in plant specialized metabolism research. The authors outline an integrated pipeline-from plant material selection to multi-omic approaches (including single-cell omics, MALDI imaging, and next-generation sequencing), cheminformatics, and validation strategies-that can facilitate the discovery of biosynthetic pathways. I have several concerns that I believe should be addressed:

Major Concerns

1. Manuscript Scope and Target Audience

It is not entirely clear whether this manuscript is intended for newcomers to the field, or if it is meant as a critical review emphasizing the importance of integrating these new technologies. Although the abstract refers to it as a "commentary," the format and content feel more akin to a methodological review. I recommend clarifying the scope and aim at the outset. For example, specify whether the manuscript aims to provide general guidance on introducing new methods (such as MALDI imaging and single-cell omics) or if it is intended as a broader, high-level commentary on the state of the field.

2. Abstract Focus

If the goal is to provide a general introduction and methodological guidance, the abstract may be slightly misleading. It emphasizes incorporating new technology into existing workflows, but it does not clearly state whether the manuscript is a broad commentary or a detailed methodological review. A more precise statement of purpose in the abstract would help set the reader's expectations.

3. Detail on Highlighted Technologies

Although single-cell omics and MALDI imaging are emphasized in the abstract, the text lacks detailed discussion on these techniques. It would be helpful to include specifics on how these methods can address major challenges in plant specialized metabolism research, as well as potential pitfalls and practical considerations.

Minor Concerns

- Line 185: Provide a brief explanation of the Hi-C abbreviation for clarity.
- Line 129: While I agree with the point on Nagoya Protocol compliance, it should also be noted that implementing the protocol remains challenging in many countries, where established processes are lacking. Offering practical guidelines or resources would be especially beneficial for readers unfamiliar with these requirements.
- Line 167: From a practical standpoint, how many tissues or organs should be collected and sequenced? Additionally, for reference transcriptome assembly, Iso-Seq may be a suitable approach to capture full-length transcripts.
- Line 175: Preparing protoplasts from many plant samples is extremely challenging. Single-nucleus sequencing might be more feasible in such cases.
- Line 178: Consider adding a concise introduction to single-cell omics methods, highlighting how they differ from bulk RNA-seq. A brief mention of spatial transcriptomics developments would also enrich the discussion.
- Line 211: While multi-omic data integration is powerful, it typically requires sampling multiple tissues to achieve sufficient statistical power. Including guidelines on experimental design (i.e., which tissues to target and recommended sample sizes) would be valuable for researchers planning similar studies.
- Lines 390-414: In the Future Perspectives section, it would be beneficial to highlight current bottlenecks in the field and propose potential solutions or research directions.
- Additional Suggestion: If space permits, adding a box to describe different sequencing methods (e.g., short-read vs. long-read, single-cell vs. single-nucleus) with their pros and cons would provide practical insight for readers.

Figure 1

- The flowchart is a great concept, but it appears somewhat crowded and may overwhelm readers. Consider using different colors or shapes to differentiate questions, actions, and methods. Grouping related methods or data types into submodules could also help, allowing the main flowchart to remain high-level and easier to follow.

Referee #2:

This commentary article highlights how advances in knowledge and technology can be used to identify genes and pathways for plant specialized metabolites. The article is well written and is up-to-date. Below are areas that could be improved for completeness or clarity.

-From reading the article, I could not sense what the target audience was expected to be as some areas in specialized metabolism discovery are not covered in depth yet others are covered in excruciating details such as NMR terms. Being this is an article for the EMBO Journal, I am going to assume the reader is potentially a biologist that is not a specialist in this area or a graduate student attempting to learn the breadth of the field. One suggestion is for the authors to explicitly state who this commentary article is geared towards, then make sure the sections are balanced with respect to depth/detail of information.

-The section "Augmenting natural products research with Cheminformatics" is mostly on databases and not cheminformatics methods/approaches to expand our knowledge or predict specialized metabolism. This should be re-titled or actual cheminformatics methods/approaches described.

-Line 78: Errors can also occur in that the study organism in the publication has not been properly taxonomically identified. Also, the lack of provenance/metadata, especially in older publications, and lead to mis-assumptions.

-Line 122: Also highlight that even within the same species there can be chemotypes and thus, their specimen may not mirror what was published.

-Line 165: should be focusing

-Line 233: Discuss the limitations of RNAi: not possible with all species or organs, do not typically achieve 100% silencing, transient nature of silencing

-Box 2 NMR methods: This is a lot of detail that the average reader is not going to be able to understand. Remove or distill NMR down so that the target audience can understand how it is used.

We thank the two reviewers for their expert opinions on our manuscript. We have incorporated their feedback and have addressed each of the points raised, as outlined below. In our responses below, we have highlighted changes we have made in red text.

Referee #1:

In this manuscript, Payet et al. present an extensive overview of the latest technological advances in plant specialized metabolism research. The authors outline an integrated pipeline-from plant material selection to multi-omic approaches (including single-cell omics, MALDI imaging, and next-generation sequencing), cheminformatics, and validation strategies-that can facilitate the discovery of biosynthetic pathways. I have several concerns that I believe should be addressed:

Major Concerns

1. Manuscript Scope and Target Audience

It is not entirely clear whether this manuscript is intended for newcomers to the field, or if it is meant as a critical review emphasizing the importance of integrating these new technologies. Although the abstract refers to it as a "commentary," the format and content feel more akin to a methodological review. I recommend clarifying the scope and aim at the outset. For example, specify whether the manuscript aims to provide general guidance on introducing new methods (such as MALDI imaging and single-cell omics) or if it is intended as a broader, high-level commentary on the state of the field.

We thank the reviewer for their recommendation. The abstract has been revised as follows to emphasise who the intended audience is and the aim of the article:

“Recent developments in single-cell -omic and metabolite imaging technologies and the increasing availability of high-quality genome assemblies are having a transformative impact on the way research is carried out into plant specialised metabolism. Integrating these technologies into pathway discovery projects is therefore highly advantageous. Here, we present a general introduction into methods and workflows in specialised metabolism research. We review a range of recent methodologies, highlighting what they might be used for and common pitfalls which may be encountered. Finally, we provide a practical guide on how these technologies may be incorporated into a specialised metabolic pathway discovery pipeline for researchers who are new to the field.” **Lines 27-34**

We have also added the sentence in red below to the end of the first paragraph of the Introduction:

“These steps encompass determining a product of interest or species of interest to work on, identifying candidate biosynthetic genes and pathways, and elucidating the activities of these in a heterologous host. We emphasise recent technological advancements in the specialised metabolism space and outline key practical considerations for those who are unfamiliar with the technologies, or how they may be incorporated into a gene discovery pipeline.” **Line 68 - 70**

2. Abstract Focus

If the goal is to provide a general introduction and methodological guidance, the abstract

may be slightly misleading. It emphasizes incorporating new technology into existing workflows, but it does not clearly state whether the manuscript is a broad commentary or a detailed methodological review. A more precise statement of purpose in the abstract would help set the reader's expectations.

The abstract has been re-written as above.

3. Detail on Highlighted Technologies

Although single-cell omics and MALDI imaging are emphasized in the abstract, the text lacks detailed discussion on these techniques. It would be helpful to include specifics on how these methods can address major challenges in plant specialized metabolism research, as well as potential pitfalls and practical considerations.

We thank the reviewer for their recommendation. We have expanded the detail and specifics of the MALDI-imaging and single-cell omics as follows:

MALDI-imaging

“More recently, MALDI imaging approaches (which do not involve homogenisation) have been used to investigate the localisation of target compounds at the cell type level, thus greatly improving the resolution of these endeavours. In these methods, plant material is finely sliced (e.g. to 80µm thickness (Yamamoto *et al*, 2019)) and mounted on a glass slide. The sample is then freeze-dried, immersed in a matrix solution to facilitate ionisation, and then subjected to MALDI-imaging spectrometry (Nakabayashi *et al*, 2021; Yamamoto *et al*, 2019).” **Lines 138-141**

“These approaches can also be combined with MALDI-imaging (Schwaiger-Haber *et al*, 2023), allowing for highly precise determination of sites of synthesis at the cell type level. Furthermore, knowledge of the precise localisation of a given compound in a native tissue can suggest potential function and can yield insights into how precursors are partitioned to facilitate flux.” **Lines 154-155**

“This has allowed for localisation within tissues (Nakabayashi *et al*, 2021) and even at the level of specific cell types within tissues (Yamamoto *et al*, 2019), down to a resolution of 10µm. Whilst MALDI-imaging is conceptionally straightforward, the selection of an appropriate matrix for a tissue of interest can be empirical, potentially requiring a number of different options to be explored (Aolga *et al*, 2024). Additionally, it does rely heavily on tissue sectioning, which may be difficult in tissues which are very small or difficult to work with.” **Lines 145-146**

Single-cell omics

The following discussion of single-cell / single nucleus techniques has been introduced:

“These sequencing methods rely on the dissociation of tissues into cells in suspension, followed by preparation of protoplasts (Cole *et al*, 2021). The suspended protoplasts are then attached onto individually barcoded particles using microfluidics, followed by preparation of libraries and sequencing. The data can then be resolved into distinct cell types through use of principle component clustering (Cole *et al*, 2021). It should be noted, however, that the preparation of protoplasts for some tissues and some species can be extremely challenging. In such cases, single nucleus sequencing might be more feasible. The workflow for single nucleus sequencing is similar to single cell, but crucially it does not rely on the preparation of protoplasts (Sunaga-Franze *et al*, 2021). These methods are contrasted in Box 2, and well-reviewed by Ding *et al* (2020).” **Lines 180-188**

The pros and cons of the single cell vs single nucleus approaches are now summarised in Box 2.

Minor Concerns

- Line 185: Provide a brief explanation of the Hi-C abbreviation for clarity.

This has been done:

“In addition to transcriptome sequencing applications, SMRT sequencing has had a transformative impact on genomics, resulting in a dramatic increase in the number of sequenced plant genomes. This can be combined with **chromosome conformation capture technologies (Hi-C), which reveal contacts between genomic regions which are in close 3D proximity (Simkova et al, 2024),** to resolve even very complex genomes down to chromosome level assemblies (Chávez Montes *et al*, 2022).” **Lines 208-209**

- Line 129: While I agree with the point on Nagoya Protocol compliance, it should also be noted that implementing the protocol remains challenging in many countries, where established processes are lacking. Offering practical guidelines or resources would be especially beneficial for readers unfamiliar with these requirements.

An additional sentence has been added to the relevant section (see red text below):

“Whichever route is taken, it is important to secure information about the provenance of the material in order to ensure Nagoya Protocol compliancy (Aubertin & Filoche, 2011). **This can be challenging in many countries, where established processes are lacking. Smith et al (2018) provide comprehensive guidelines on navigating Nagoya and are a good first point of call.**” **Lines 121-122**

- Line 167: From a practical standpoint, how many tissues or organs should be collected and sequenced? Additionally, for reference transcriptome assembly, Iso-Seq may be a suitable approach to capture full-length transcripts.

The text has been revised as follows:

“It is therefore preferable to generate in-house NGS resources from the accession in use. This may involve performing RNAseq on different tissues, followed by either mapping back to a published transcriptome, or more commonly generation of *de novo* transcriptome resources for different plant tissues/treatment conditions. **Typically, this would involve short-read Illumina sequencing. However, more recently long-read Iso-Seq has been employed in the construction of reference-quality transcriptomes (Zhang et al, 2022). Whilst transcriptomes can be obtained from a single library, it is advisable to design a sequencing experiment to cover a range of tissues in which biosynthesis is and is not taking place, as this allows for exclusion of genes that are unlikely to be involved in the pathway of interest. Furthermore, quantitative information is often a useful line of evidence in pathway discovery: for example, genes that have very low expression may be excluded, and genes that are differentially expressed in tissues where biosynthesis is taking place can be shortlisted. It is therefore recommended that a minimum of three replicates per tissue, preferably four, are sequenced, as this allows for robust, statistical quantification of transcript abundance.**” **Lines 167-176**

- Line 175: Preparing protoplasts from many plant samples is extremely challenging. Single-nucleus sequencing might be more feasible in such cases.

We have combined our response to this with the point raised below.

- Line 178: Consider adding a concise introduction to single-cell omics methods, highlighting how they differ from bulk RNA-seq. A brief mention of spatial transcriptomics developments would also enrich the discussion.

“Single-cell sequencing technologies have also recently been developed, and have considerable potential for facilitating elucidation of natural product pathways that are restricted to specific cell types (Lin *et al*, 2023; Wang *et al*, 2022), as well as examples of pathways which are split between multiple, different cell types (Ozber & Facchini, 2022; Sun *et al*, 2023). These sequencing methods rely on the dissociation of tissues into cells in suspension, followed by preparation of protoplasts (Cole *et al*, 2021). The suspended protoplasts are then attached onto individually barcoded particles using microfluidics, followed by preparation of libraries and sequencing. The data can then be resolved into distinct cell types through use of principle component clustering (Cole *et al*, 2021). It should be noted, however, that the preparation of protoplasts for some tissues and some species can be extremely challenging. In such cases, single nucleus sequencing might be more feasible. The workflow for single nucleus sequencing is similar to single cell, but crucially it does not rely on the preparation of protoplasts (Sunaga-Franze *et al*, 2021). These methods are contrasted in Box 2, and well-reviewed by Ding *et al* (2020). These single cell and nucleus sequencing methods are substantially different from bulk RNA-seq, in which whole tissues are homogenised, as the information on which cells are expressing which genes is lost. Furthermore, transcript abundance in bulk RNA-seq is a synthesis of level of expression and abundance of cell type in which expression is taking place. Single-cell sequencing circumvents this problem, however in both methods the architecture of where genes are expressed in the context of a tissue is lost. Very recently, spatial transcriptomic methods such as stereo-seq have been developed and incorporated into plant science, which can overcome both these issues (Yin *et al*, 2023)” **Lines 180-195**

- Line 211: While multi-omic data integration is powerful, it typically requires sampling multiple tissues to achieve sufficient statistical power. Including guidelines on experimental design (i.e., which tissues to target and recommended sample sizes) would be valuable for researchers planning similar studies.

The text has been revised as follows:

“These multi-omic methods have historically been applied to microbial systems but are successfully being implemented in plant systems (Li *et al*, 2023). These approaches are particularly powerful, as they incorporate multiple lines of evidence. It should be noted, however, that to achieve sufficient statistical power it is generally necessary to sample multiple tissues/conditions in replicate, with a minimum of three, preferably four replicates per sample; however this can be expensive. Moreover, such an experiment should be designed to span both tissues/conditions where biosynthesis is known or suspected and tissues/conditions where it is not, so that meaningful comparisons can be made.” **Lines 239-243**

- Lines 390-414: In the Future Perspectives section, it would be beneficial to highlight current bottlenecks in the field and propose potential solutions or research directions.

A new paragraph has been added to the beginning of the Future perspectives section as follows:

“Developments in NGS methods and metabolite imaging have greatly enriched the capacity for researchers to identify putative biosynthesis genes from a wide variety of non-reference organisms. However, one of the key bottlenecks in natural product discovery remains the ratification of biosynthesis genes. Increasing the accuracy of prediction of biosynthesis genes will therefore be vital in overcoming this. In the future, the development of large databases of ratified functional genes that can be used to train machine learning algorithms will greatly accelerate this capability.” **Lines 352-357**

- Additional Suggestion: If space permits, adding a box to describe different sequencing methods (e.g., short-read vs. long-read, single-cell vs. single-nucleus) with their pros and cons would provide practical insight for readers.

A new box (Box 2) contrasting these technologies has been introduced.

Figure 1

- The flowchart is a great concept, but it appears somewhat crowded and may overwhelm readers. Consider using different colors or shapes to differentiate questions, actions, and methods. Grouping related methods or data types into submodules could also help, allowing the main flowchart to remain high-level and easier to follow.

The figure has been revised in response to the reviewer’s comments.

We thank reviewer 1 for their detailed feedback. We feel that incorporation of their points has greatly improved our manuscript.

Referee #2:

This commentary article highlights how advances in knowledge and technology can be used to identify genes and pathways for plant specialized metabolites. The article is well written and is up-to-date. Below are areas that could be improved for completeness or clarity.

We thank Reviewer 2 for their very positive comments.

-From reading the article, I could not sense what the target audience was expected to be as some areas in specialized metabolism discovery are not covered in depth yet others are covered in excruciating details such as NMR terms. Being this is an article for the EMBO Journal, I am going to assume the reader is potentially a biologist that is not a specialist in this area or a graduate student attempting to learn the breadth of the field. One suggestion is for the authors to explicitly state who this commentary article is geared towards, then make sure the sections are balanced with respect to depth/detail of information.

The abstract has been rewritten, and additional lines have been added to the introduction to clarify the target audience (see response to Reviewer 1, comments 1 and 2). To address depth/detail balancing, we have done the following:

The text below was removed to cut down on detail and make the article more accessible to the intended audience.

~~“For example, triterpenes containing a double bond typically undergo a reverse Diels-Alder reaction upon subjection to the electron impact ionisation method commonly used in GC-MS systems, leading to two characteristic fragment ions ($m/z = 218$ and 279 for β -amyrin based triterpenes when derivatised with trimethyl silyl groups) (Burnouf-Radosevich *et al*, 1985; Uddin *et al*, 2022). A shift in the m/z values of these fragments can identify the specific ring of the triterpene bearing a modification. Similarly, fragmentation of molecules containing sugar chains, such as saponins, in an electrospray chamber will lead to the loss of the sugars in sequence while usually leaving the aglycone intact, creating a series of fragments each losing a set mass equal to a specific sugar, which allows the determination of the identities of the attached sugars as well as providing clues to the order of sugars in a chain (Jo *et al*, 2024).”~~

Lines X-Y of the original have been replaced with the following to remove extraneous detail:

~~“More recently, the combined use of HSQC-TOCSY (heteronuclear single quantum correlation-total correlation spectroscopy) and heteronuclear two-bond correlation (H2BC) has proven beneficial for assigning carbon and proton resonances within sugar moieties (Graziani *et al*, 2018; Shiomi *et al*, 2016; Wallace *et al*, 2022).”~~ **Lines 345-348**

~~-The section "Augmenting natural products research with Cheminformatics" is mostly on databases and not cheminformatics methods/approaches to expand our knowledge or predict specialized metabolism. This should be re-titled or actual cheminformatics methods/approaches described.~~

~~This section has been renamed to “Augmenting natural products research through the use of cheminformatics databases.”~~

~~-Line 78: Errors can also occur in that the study organism in the publication has not been properly taxonomically identified. Also, the lack of provenance/metadata, especially in older publications, and lead to mis-assumptions.~~

The following text (highlighted in red) has been added to address this point:

~~“Additionally, cross-referencing with taxonomic databases such as the National Center for Biotechnology Information (NCBI) taxonomy database (Schoch *et al*, 2020) can help confirm the accuracy of species names. Errors may also arise in cases in which the organism of study has not been properly taxonomically identified. This is particularly difficult to resolve in studies which lack sufficient metadata or information on provenance, as it can lead to misassumptions.”~~ **Lines 100-103**

The text immediately after this has also been revised from:

~~“The curated and filtered data can be valuable for various purposes”~~ to ~~“Curated and filtered data from these databases can be valuable for various purposes”~~. **Line 104**

~~-Line 122: Also highlight that even within the same species there can be chemotypes and thus, their specimen may not mirror what was published.~~

A sentence to this effect has been included:

~~“The metabolite profiles of commercially purchased plant material may differ substantially from *bona fide* botanic garden accessions, possibly due to nomenclature issues or hybridisation with other closely related species. Indeed, even within the same species it is well documented that different chemotypes can exist between individuals, and thus the~~

specimen of study may not mirror published data (Anaia *et al*, 2024; Ziaja & Müller, 2025).“
Lines 125-127

-Line 165: should be focusing

Changed to focusing

-Line 233: Discuss the limitations of RNAi: not possible with all species or organs, do not typically achieve 100% silencing, transient nature of silencing

A sentence to address this has been included as follows:

“In this approach, small interfering RNAs (siRNA) can be expressed to activate endogenous gene silencing mechanisms, thereby reducing expression of the candidate gene and, correspondingly, production of the target molecule (Boccia *et al*, 2024; Jo *et al*, 2024). Whilst this is an effective strategy, it should be noted that typically RNAi methods do not achieve complete silencing, and so this method rarely results in a complete knockout phenotype. Furthermore, silencing is often transient, and over time the efficacy may reduce.” **Lines 280-282**

-Box 2 NMR methods: This is a lot of detail that the average reader is not going to be able to understand. Remove or distill NMR down so that the target audience can understand how it is used.

Box 2 (now Box 3) has been revised to increase clarity. Axis information was removed, the ROESY section was incorporated into the NOESY section, and extraneous details were purged from the HSQC section.

Following input from the reviewers, Box 4 (Take home messages) has been modified as below:

Box 4 Take home messages

- Cheminformatics databases can be used to systematically identify compounds based on structural trends (e.g. shared aglycone) and bioactivity and can also be integrated with taxonomy to investigate chemistry across genera.
- Understanding where and when biosynthesis occurs is vital for metabolic pathway elucidation.
- Single cell omics techniques are powerful additions to pathway elucidation, but it must be taken into consideration that some pathways are segmented over a variety of different cell types.
- Heterologous reconstitution of biosynthesis pathways in *Nicotiana benthamiana* is effective and highly scalable.

Dear Rocky,

I am pleased to inform you that your manuscript has been accepted for publication in the EMBO Journal.

Congratulations! I'm sure this will become a valuable reference.

Yours sincerely,

William

William Teale, PhD
Editor
The EMBO Journal
w.teale@embojournal.org
